# Nosustrophine: An Epinutraceutical Bioproduct with Effects on DNA Methylation, Histone Acetylation and Sirtuin Expression in Alzheimer’s Disease

**DOI:** 10.3390/pharmaceutics14112447

**Published:** 2022-11-12

**Authors:** Olaia Martínez-Iglesias, Vinogran Naidoo, Iván Carrera, Lola Corzo, Ramón Cacabelos

**Affiliations:** EuroEspes Biomedical Research Center, International Center of Neuroscience and Genomic Medicine, 15165 Bergondo, Corunna, Spain

**Keywords:** Alzheimer’s disease, DNA methylation, epinutraceutical, HDAC, PSEN2, Sirtuin, 3xTg-AD

## Abstract

Alzheimer’s disease (AD), the most common cause of dementia, causes irreversible memory loss and cognitive deficits. Current AD drugs do not significantly improve cognitive function or cure the disease. Novel bioproducts are promising options for treating a variety of diseases, including neurodegenerative disorders. Targeting the epigenetic apparatus with bioactive compounds (epidrugs) may aid AD prevention treatment. The aims of this study were to determine the composition of a porcine brain-derived extract Nosustrophine, and whether treating young and older trigenic AD mice produced targeted epigenetic and neuroprotective effects against neurodegeneration. Nosustrophine regulated AD-related *APOE* and *PSEN2* gene expression in young and older APP/BIN1/COPS5 mice, inflammation-related (*NOS3* and *COX-2*) gene expression in 3–4-month-old mice only, global (5mC)- and de novo DNA methylation (*DNMT3a*), *HDAC3* expression and HDAC activity in 3–4-month-old mice; and *SIRT1* expression and acetylated histone H3 protein levels in 8–9-month-old mice. Mass spectrometric analysis of Nosustrophine extracts revealed the presence of adenosylhomocysteinase, an enzyme implicated in DNA methylation, and nicotinamide phosphoribosyltransferase, which produces the NAD+ precursor, enhancing SIRT1 activity. Our findings show that Nosustrophine exerts substantial epigenetic effects against AD-related neurodegeneration and establishes Nosustrophine as a novel nutraceutical bioproduct with epigenetic properties (epinutraceutical) that may be therapeutically effective for prevention and early treatment for AD-related neurodegeneration.

## 1. Introduction

Alzheimer’s disease (AD) is the most common cause of dementia and a high-priority public health issue. It is approximately 2–3 more prevalent in women than in men [1] and is expected to affect 130 million people in 2050 [2]. AD is progressive and causes permanent memory loss and impairment of cognitive function [3]. The presence of extracellular amyloid-beta (Aβ) plaques, intracellular neurofibrillary tangles caused by the accumulation of hyperphosphorylated microtubule-associated tau protein, and neuroinflammation, are the primary pathogenic hallmarks of AD [4].

In terms of its cognitive profile, AD is defined mainly by deficits in episodic memory because of damage to the temporal lobe. However, other brain areas may also contribute to the clinical picture in AD. Dopaminergic neurons in the ventral tegmental area (VTA) in the brainstem, for example, play critical roles in reward- and goal-directed behaviors [5], and project to several targets such as the hippocampus [6] and the nucleus accumbens [7]. Interestingly, there is a selective degeneration of VTA dopamine neurons during the pre-plaque stages in the Tg2576 transgenic mouse model of AD [8]. Furthermore, there is an association between altered VTA connectivity and volume versus memory impairment in patients with early (prodromal) AD [9].

The most common pharmacological interventions for AD are natural products (25.6%), anti-Aβ compounds (13.3%), neurotransmitter enhancers (11.4%), multi-target- (2.5%) and anti-tau drugs (2.3%) [10]. However, most studies focus on the acetylcholinesterase (AChE) inhibitors Rivastigmine, Galantamine and Donepezil, the N-Methyl-D-Aspartate receptor antagonist Memantine, and combination treatments. While these agents delay AD progression and symptom development, they do not substantially enhance cognitive function or cure AD [11]. Moreover, these therapies produce many adverse effects including organ toxicity [12]. A considerable body of research supports the advantages of novel therapeutic compounds, derived from vegetal, marine, or animal bioderivatives in nature, for delaying or preventing neuropathology-induced cognitive decline and other age-related disorders [12,13,14]. Curcumin, resveratrol and genistein, for example, inhibit Aβ peptide formation, aggregation and deposition [12,14]; these compounds also function as epigenetic modulators for preventing and treating various diseases [14,15]. Epigenetics refers to reversible heritable alterations in gene expression without modifications to the DNA sequence, linking the genome with the surrounding environment [16,17,18]. DNA methylation, chromatin remodeling/histone modifications and micro RNA (miRNA) regulation are the classic epigenetic mechanisms [16,18]. Epigenomic dysregulation may contribute to the etiology and progression of neurodegenerative disorders [19]. Global DNA methylation (5mC) levels are substantially altered in the brain during the early stages of AD and may contribute to the cognitive deficits observed in patients with AD. Furthermore, changes in DNA methylation have been discovered in AD-related genes; some of these genes are hypomethylated (e.g., *APP*, *BACE*, and *PSEN1*), whereas others (e.g., *APOE* and *MAPT*) are hypermethylated [20]. In addition to DNA modifications, histone acetylation is dysregulated in AD, with negative consequences on cognitive function. Preclinical data show that histone deacetylase (HDAC) inhibitors increase cognitive ability, at least in the short term. During aging, sirtuins (SIRTs) help maintain neuronal functions, but also regulate AD-linked mechanisms such as neuroinflammation, oxidative stress, and processing of tau protein and amyloid precursor protein (APP). Low levels of SIRT1 in the AD brain is associated with Aβ and tau accumulation, and decreased SIRT1 concentrations enhance the levels of hyperphosphorylated tau. SIRT1 overexpression is neuroprotective in AD, possibly through promoting cellular proteostasis and neurotrophic factor activation [21].

Nootropics, compounds that improve concentration, motivation, memory and attention [22], may help strengthen cognitive function in individuals with neurodegenerative diseases [23]. Promising results have been obtained in clinical trials of AD with the nootropics *Crocus sativus* (saffron extract) in treating mild-to moderate AD [24], phospholipase A2 for treating early stage AD [25], and the omega-3 fatty acid docosahexaenoic acid, which promotes verbal recognition memory in patients with age-related cognitive decline [26]. Cerebrolysin, a porcine brain-derived peptidergic preparation with pharmacodynamic properties similar to endogenous neurotrophic factors, is another nootropic-of-interest. That drug is reported to contain enkephalins, orexin, and the neurotrophins brain-derived neurotrophic factor (BDNF), ciliary neurotrophic factor (CNTF), and nerve growth factor (NGF) [27], but this might not be the case [28]; the latter study revealed an absence of these factors in Cerebrolysin. In AD pathogenesis, there is progressive dysregulation of neurotrophin synthesis, release and function [29], including aberrant BDNF/tropomyosin receptor kinase (Trk)B and neurotrophin-3 (NT-3)/TrkC signaling [30], which highlight their importance as potential treatments/targets for alleviating AD-related neuropathology. For example, the ginkgo flavonols isorhamnetin, quercetin and kaempferol stimulate cAMP-response element binding protein (CREB)-BDNF signaling, decrease hippocampal Aβ production and improve spatial learning in APP/presenilin 1 (APP/PS1) transgenic mice [31]. However, clinical therapeutic application of recombinant neurotrophic factors is hampered by barriers such as undesirable pharmacokinetic features, low blood-brain barrier permeability, and significant side effects [32].

Based on these findings, we hypothesized that a nutraceutical bioproduct with epigenetic properties (epinutraceutical) may correct AD-induced epigenetic changes, thus presenting an additional therapeutic approach for treating AD. Nosustrophine is a porcine-derived brain extract manufactured through non-denaturing biotechnological processes.

The aims of the present study were to (i) determine the composition of Nosustrophine, and (ii) analyze, using a transgenic (APP/BIN1/COPS5) mouse model of AD, whether Nosustrophine acts as an epinutraceutical by modulating the epigenetic machinery that promotes AD progression. Here, we analyzed the effects of Nosustrophine on neurodegeneration-related gene expression, DNA methylation and post-translational histone modifications. The pharmacoepigenetic response of these epigenetic biomarkers to Nosustrophine therapy in vivo suggests that it may be considered an epinutraceutical bioproduct for treating AD. 

## 2. Materials and Methods

### 2.1. Nutritional Analysis of Nosustrophine

Nosustrophine is a nutraceutical bioproduct that was developed through non-denaturing biotechnological processes from the domestic pig (*Sus scrofa domesticus*) brain (Patent ID: P202230047/ES2547.5). To analyze the nutritional composition (fatty acids, amino acids, vitamins, nutritional minerals, heavy metals, and other compounds) of Nosustrophine, chromatography, spectroscopy, spectrometry methods, and radioimmunoassay (RIA) and enzyme-linked immunosorbent assays (ELISAs) were conducted at Alkemi (AGQ Lab Group, Madrid, Spain) (Figure 1). Fatty acid content in the extract was quantified by gas chromatography with flame-ionization detection (GC-FID). To quantify carbohydrate (fructose, galactose, glycerol, glucose, lactitol, lactose, maltitol, maltose, mannitol, sucrose and sorbitol), amino acid, and vitamin (B-complex, C, D and E) content, high performance liquid chromatography (HPLC) with ultraviolet-visible (UV-Vis) detection was used. Nutritional mineral content was determined by flame atomic absorption spectrometry (FAAS), with the exception of calcium, which was quantified using HPLC UV-Vis spectrophotometry. All general chemicals were purchased from Sigma-Aldrich (St. Louis, MO, USA) unless otherwise indicated.

### 2.2. Catecholamine, Serotonin and L-DOPA Analyses

Fractionated catecholamines and serotonin were measured in Nosustrophine extracts by ultra-high-performance liquid chromatography (UHPLC) with electrochemical detection (ECD) in the ALEXYS Neurotransmitter Analyzer (Antec Scientific, Boston, MA, USA). Prior to HPLC analysis, the powdered extract was dissolved in phosphate-buffered saline (PBS, pH 7.4) and mixed by vortexing. The solution was then centrifuged at 4000 rpm and the supernatant collected for analysis. A sample clean-up was performed using a solid phase extraction for catecholamine analysis. In the case of serotonin, a precipitation step was applied to remove the sample matrix and to spike the sample with an internal standard. The concentration of histamine in the Nosustrophine extract was measured by HPLC with fluorescence detection (FD). The chromatographic system comprised four independent isocratic pumps, a stainless-steel column with a cation exchanger (TSK gel SP-2SW, 5 μm; TosoHaas Corporation, Montgomeryville, PA, USA) and a fluorometric detection system. Supernatants (20 µL) were injected directly into the HPLC column. This method used perchloric acid for histamine extraction, followed by direct HPLC analysis with on-line derivatization with o-phthaldialdehyde and FD, at excitation and emission wavelengths of 360 nm and 450 nm, respectively. 

To measure the concentration of L-DOPA, the supernatants obtained from Nosustrophine extracts were diluted in 5% perchloric acid, and subsequently filtered through a nylon membrane filter (Dismic-13NP, 0.45 μm pore size, 13 mm diameter, Advantec, Tokyo, Japan), according to the method by Burbano et al. [33]. As these samples were thermosensitive, they were kept on ice throughout the extraction process. L-DOPA levels were then measured in these samples by HPLC-UV with a Varian 920-LC analyzer (Agilent Technologies, Santa Clara County, CA, USA). L-DOPA analysis was performed by URIKER laboratory (Bizkaia, Spain).

### 2.3. Neurotrophic Factor Analysis

Concentrations of BDNF and corticotropin-releasing factor were quantified with ELISAs as per the manufacturer’s guidelines. Somatostatin concentrations were measured by RIA. Data are expressed in pg/mg.

### 2.4. Proteomic Analysis

#### 2.4.1. Sample Preparation

For proteomic analysis, a sample of the Nosustrophine extract was sent to the Proteomics Unit at the Instituto de Investigación Biomédica de A Coruña (INIBIC) (Spain) (Appendix A). Lyophilized (1 mg) sample extracts of Nosustrophine were subjected to LC-MS-based proteomic analysis at INIBIC (A Coruña). The samples were first homogenized with lysis buffer (100 µL) (2% SDS, sodium dodecyl sulfate/6 M urea/25 mM ammonium bicarbonate, Ambi), vortexed thoroughly until a homogeneous solution was obtained, and then bath-sonicated for three minutes. To eliminate cell debris and other insoluble material, the extracts were centrifuged at 13,000 rpm at 4 °C for 15 min. Proteins were then precipitated with acetone at −20 °C overnight and resuspended in 6 M urea/2 M thiourea/25 mM Ambi. Protein concentrations were determined with a Bradford assay and the absorbance measured at 590 nm in a spectrophotometer, against a bovine serum albumin standard curve. SDS-polyacrylamide gel electrophoresis (PAGE) (1 µg protein/lane) followed by silver nitrate staining were used to confirm sample quality. 

#### 2.4.2. Liquid Chromatography-Tandem Mass Spectrometry (LC-MS/MS) Analysis

Equal quantities (20 µg) of each sample were reduced with 10 mM dithiothreitol for 1 h at 37 °C and then alkylated with 50 mM iodoacetamide at room temperature for 45 min in the dark. The samples were then digested for 16 h at 37 °C with sequencing-grade modified trypsin (Promega, Madrid, Spain) at an enzyme-to-substrate ratio of 1:40, and treated with 10% trifluoroacetic acid until a pH value of 3 was reached. The digested peptides were desalted using stage-tips that were manufactured in-house (3M Empore SPE-C18 disk, 47 mm, Sigma-Aldrich) and then speed-vacuum-dried (Thermo Fisher, Waltham, MA, USA). The dried eluates were then reconstituted in 2% acetonitrile (ACN) and 0.1% formic acid (FA) diluted in water for direct LC-MS. The peptide mixture (200 ng) was added to a nanoElute (Bruker Daltonics) nano-flow LC-coupled to a high-resolution trapped ion mobility spectrometry (TIMS) on a quadrupole TOF (Q-TOF) (timsTOF Pro, Bruker Daltonics) with a CaptiveSpray ion source (Bruker Daltonics). LC was performed at 50 °C and with a constant flow of 500 nL/min on a reversed-phase column (15 cm × 75 m i.d.) with a pulled emitter tip, packed with 1.9 m C18-coated porous silica beads (Dr. Maisch, Ammerbuch-Entringen, Germany). Chromatographic separation was performed for 40 min with a linear gradient of 5–35% buffer B (100% ACN and 0.1% FA). Following electrospray ionization (ESI), the peptides were analyzed in data-dependent scan mode with Parallel Accumulation-Serial Fragmentation (PASEF) enabled.

#### 2.4.3. LC-MS/MS Data Analysis

PEAKS Studio 10.6 build 20201221 (Bioinformatics Solutions Inc., Waterloo, ON, Canada) was used to process the raw mass spectrometry data. The MS/MS spectra were matched to in silico-derived fragment mass values of tryptic peptides against Pig UniProtKB/Swiss-Prot (version 2022_02). The search parameters were: Parent Mass Error Tolerance: 15.0 ppm; Fragment Mass Error Tolerance: 0.05 Da; Enzyme: Trypsin; Fixed Modifications: Carbamidomethylation; Variable Modifications: acetylation (Protein N-terminus), deamidation (NQ), oxidation (M) and acetylation (N-terminus); Max Variable PTM Per Peptide: 3. Matches were filtered for 1% FDR at the peptide level. The Spec value, based on peptide spectrum matches (PSM), was used to determine the relative abundance of the proteins in each sample. The proteins identified with less than two unique peptides were excluded from the analysis. To facilitate the overall analysis, protein bioinformatic classification was performed with the PANTHER Classification System.

### 2.5. Animals and Genotyping

All experiments were conducted in adherence to European Community Law (86/609/EEC), European Union Directive 2016/63/EU and the Spanish Royal Decree (R.D. 1201/2005). The Ethics Committee of the International Center of Neuroscience and Genomic Medicine approved the procedures performed in this study. The experimental design for the in vivo experiments is shown in Appendix A.

Wild-type C57BL/6 and APP/BIN1/COPS5 3xTg-AD mice were used from colonies previously supplied by Dr. M.K. Lakshmana (Alzheimer’s Disease Research Unit, Herbert Wertheim College of Medicine, Florida International University, Miami, FL, USA). These mice overexpress the Swedish mutation of human APP, bridging integrator 1 (BIN1) and COP9 constitutive photomorphogenic homolog subunit 5 (COPS5), C57BL/6 mice were the background strain. Mice were housed in an air-conditioned room (50–60% humidity; 22 ± 0.5 °C;) under a 12 h light:12 h dark cycle with unrestricted access to water and food. 

Genomic DNA from tail samples (as templates) was used to genotype mice with the polymerase chain reaction (PCR). The High Pure PCR Template Preparation kit (Roche) was then used to extract genomic DNA, according to the manufacturer’s specifications. To amplify *APP* and *BIN*/*COPS5*, the UltraRun LongRange PCR kit (Qiagen, Hilden, Germany) and Phire Hot Start II DNA polymerase (Thermo Fisher Scientific, Waltham, MA, USA) were used, respectively. The primers for genotyping are listed in Appendix A, and the experimental conditions for PCR are shown in Appendix A. For DNA extraction and amplification, IL-2 primers were used as controls.

### 2.6. Preparation of Nosustrophine and Treatments In Vivo

A stock solution of lyophilized Nosustrophine (20 mg/mL) was prepared using sterile-filtered 0.9% NaCl, and sonicated on ice. Wild-type and APP/BIN1/COPS5 3xTg-AD mice (3–4-month-old and 8–9-month-old mice) were randomized into groups and injected i.p. daily for four weeks with saline (vehicle) or Nosustrophine (100 mg/kg) (Appendix A). Four animals per experimental group were used. In terms of life phase equivalencies for mice versus humans, 3–6-month-old and 10–14-month-old mice are considered equivalent to mature- (20–30-year-old) and middle-aged (38–47-year-old) human adults, respectively [34]. In our study, we refer to 3–4-month-old mice as young, and 8–9-month-old mice as older. 

The mice were placed under anesthesia with diethyl ether (Panreac, Germany) and then subjected to transcardiac perfusion with PBS (pH 7.4). The left cerebral hemispheres were extracted, placed in 4% paraformaldehyde in 0.1 M phosphate buffer (pH 7.4) for 48 h at 4 °C, and cryoprotected by immersion in 30% sucrose in 0.1 M phosphate buffer. A cryostat (Starlet 2212, Bright; UK) was then used to acquire a parallel series of transverse sections (40–50 µm); these sections were subsequently mounted on Superfrost Plus (Menzel-Gläser, Braunschweig, Germany) slides in preparation for immunohistochemistry.

Hippocampi were separated from the right hemispheres and immersed in RNA later solution (Qiagen, Hilden, Germany) for RNA extraction, or frozen at −80 °C until DNA and nuclear extraction, Western- and dot blotting.

### 2.7. Immunofluorescence

Brain sections (*n* = 3–4 animals/group) were washed twice for 10 min each in PBS and incubated in blocking solution (15% normal goat serum in 0.1 M PBS/0.2% Tween-20) for 1 h at room temperature (RT). The sections were subsequently incubated with primary antibodies against neuronal nuclear protein (NeuN) and a marker of dopaminergic neurons (tyrosine hydroxylase, TH) (Appendix A) in blocking solution overnight at 4 °C. Negative primary controls included the blocking solution only. The tissues were washed in PBS and then incubated for 2 h at RT with a secondary antibody (Alexa-488-conjugated; Thermo Fisher, Scientific, Waltham, MA, USA) diluted in blocking solution. The sections were then rinsed three times for 15 min in PBS, and coverslipped with Vectashield medium (Vector Labs, Newark, CA, USA). Immunolabeled images were captured with a Leica DM6 B upright microscope (Leica Microsystems, Buffalo Grove, IL, USA) and Leica Application Suite X (LAS X) software. In each tissue section, the areas occupied by NeuN and TH staining (area fractions; binary area/measured area) in the CA1 subfield and dentate gyrus (DG) of the hippocampus and in the VTA, respectively, were computed with ImageJ (NIH, Bethesda, MD, USA) from three acquired z-stack datasets per antigen.

### 2.8. Western Blotting and Dot Blot Characterization

Mouse hippocampal samples were probe-sonicated (Model 100 Sonic Dismembrator, Fisher Scientific) on ice at 20% amplitude for 10 s in RIPA buffer (0.15 M NaCl; 1% Triton X-100; 0.5% sodium deoxycholate; 0.1% SDS; 50 mM Tris, pH 8.0) with protease and phosphatase inhibitors (Sigma, Kawasaki, Japan). The protein concentrations of these samples were quantified with the bicinchoninic acid (BCA) assay (Thermo Fisher Scientific, Waltham, MA, USA). For Western blot analysis, equal amounts of protein were denatured at 70 °C for 10 min, loaded in 4–12% NuPAGE Bis-Tris gels (Thermo Fisher, Waltham, MA, USA), separated at 200 V for 30 min, and then transferred to Amersham Protran 0.2 µm-pore size nitrocellulose membranes (Cytiva, Freiburg, Germany). The membranes were blocked in 5% non-fat dry milk for 1 h at room temperature and then incubated overnight at 4 °C with primary antibodies against acetyl-histone H3 (Lys14) (Merck) and histone H3 (Thermo Fisher, Waltham, MA, USA) (Appendix A). The membranes were washed in tris-buffered saline (pH 7.6), incubated in horseradish peroxidase (HRP)-conjugated donkey anti-rabbit secondary antibody (Thermo Fisher, Waltham, MA, USA) for 1 h at room temperature, and then incubated in SuperSignal West Pico Plus chemiluminescent substrate (Thermo Fisher, Waltham, MA, USA) for 5 min. Protein bands were visualized using the C-DiGit blot scanner (LI-COR, Lincoln, Dearborn, NE, USA) and densitometry analyses were performed with Image Studio (software version 5.2) (LI-COR). 

For dot blot analysis, equal amounts of hippocampal protein diluted in RIPA buffer were spotted 1 cm apart on a nitrocellulose membrane (Amersham Protran, Cytiva, Buckinghamshire, UK). Human amyloid β-peptide (1-42) (Tocris, Bio-Techne, Wiesbaden, Germany) and bovine serum albumin (Sigma, Kawasaki, Japan) were used as positive and negative controls, respectively. The membrane was air-dried for 30 min, stained with Ponceau S (Sigma, Kawasaki, Japan), and washed twice in TBS-0.1% Tween-20. The membrane was blocked in 5% non-fat dry milk for 1 h at room temperature, and then incubated in mouse Aβ1-42 (clone 12F4) (Merck) primary antibody at 4 °C overnight. The membrane was washed in TBS-0.1% Tween-20, and then incubated with a donkey anti-mouse HRP-conjugated secondary antibody (Thermo Fisher, Waltham, MA, USA) for 1 h at room temperature, and exposed to SuperSignal West Pico Plus chemiluminescent substrate for 5 min. Immunoreactive dots were visualized with the C-DiGit blot scanner. Integrated optical densities were computed with Image Studio, and normalized to the total protein per dot measured previously on the same (Ponceau-stained) membrane. 

### 2.9. Quantification of Global DNA Methylation (5mC)

DNA was extracted from mice hippocampi with the Qiagen DNA Mini Kit (Qiagen, Hilden, Germany), and only DNA samples with 260/280 and 260/230 ratios greater than 1.8. were used. Global 5mC levels were measured colorimetrically using 100 ng DNA per sample with the MethylFlash Methylated DNA Quantification Kit (Epigentek, New York, NY, USA). Absorbance was measured at 450 nm with a microplate reader (Epoch, BioTek Instruments, Winooski, VT, USA). As per the manufacturer’s guidelines, 5mC levels were expressed as mean (%) ± S.E.M. To quantify the absolute amount of methylated DNA, we generated a standard curve using linear regression (Microsoft Excel). The amount and percentage of 5mC was then calculated with the following two formulae, respectively:5mC (ng) = (Sample OD − Blank OD)/(Slope × 2)(1)
5mC (%) = 5mC (ng)/sample DNA (ng) × 100(2)

### 2.10. Quantitative Real Time RT-PCR (qPCR)

Total hippocampal RNA was extracted using the RNeasy Mini Kit (Qiagen, Hilden, Germany) as per the manufacturer’s specifications. RNA concentration and quality were then measured with a spectrophotometer (Epoch, BioTek instruments, Winooski, VT, USA). In this study, only RNA samples with 260/280 and 260/230 ratios over 1.8 were used. Purified RNAs (400 ng) were reverse-transcribed into DNA (High Capacity cDNA Reverse Transcription Kit; Thermo Fisher Scientific, Waltham, MA, USA) under the following conditions: 25 °C (10 min), 37 °C (120 min) and 85 °C (5 min). 

QPCR, with the aid of the StepOne Plus Real Time PCR system (Thermo Fisher Scientific, Waltham, MA, USA), was used to quantify gene expression. Each PCR reaction was done in duplicate using the TaqMan Gene Expression Master Mix (Thermo Fisher Scientific, Waltham, MA, USA) and TaqMan probes (Thermo Fisher Scientific, Waltham, MA, USA) (Appendix A). Relative quantification was performed with the comparative CT method with the StepOne Plus Real Time PCR software and the results are expressed as fold-induction with respect to healthy samples. Data were normalized to mouse S18 mRNA levels (Appendix A), and presented as means ± S.E.M.

### 2.11. Nuclear Protein Extraction

The EpiQuik Nuclear extraction kit (EpiGentek) was used to isolate nuclear proteins in hippocampal samples and followed the manufacturer’s guidelines. Protein concentrations were quantified with the Pierce Coomassie (Bradford) Assay Kit (Life Technology, Carlsbad, CA, USA). Absorbances were measured in a spectrophotometer (BioTek Instruments, Winooski, VT, USA) at 595 nm.

### 2.12. Quantification of HDAC and Sirtuin Activity

HDAC and sirtuin activities were assessed in mouse hippocampal samples using HDAC- or Sirtuin Activity/Inhibition colorimetric kits (EpiGentek, New York, NY, USA), respectively, following the manufacturer’s directions. Briefly, hippocampal nuclear protein extracts (50 ng) were added to wells containing an acetylated histone-derived substrate. The plate was then incubated for 90 min at 37 °C. After rinsing the wells with wash buffer, capture and detection antibodies were added. The quantities of deacetylated product, proportional to HDAC and SIRT enzyme activities, were measured at 450 nm in a microplate reader. As a reference, absorbance measurements at 655 nm were taken.

### 2.13. Statistical Analysis

Proteomic data were quantified and analyzed using PEAKS Studio 10.6 (Bioinformatics Solutions Inc., Waterloo, ON, Canada) and the statistical software ProteinPilot (AB Sciex, Foster City, CA, USA). The results obtained were then exported into Microsoft Excel for further analyses. 

For mouse studies, the Levene’s- and Shapiro-Wilk tests were used to assess equality of variances and normality, respectively. Statistical significance was determined with unpaired *t* tests or one-way analysis of variance (ANOVA) (Prism, GraphPad Software, San Diego, CA, USA) with Tukey’s post hoc tests. Data are presented as means ± S.E.M; *p* values < 0.05 were considered statistically significant.

## 3. Results

### 3.1. Screen for the Composition of Nosustrophine

We first investigated the nutritional composition of Nosustrophine using chromatographic (HPLC-UV, GC-FID, UHPLC-ECD), spectroscopic (GF-AAS, FAAS), and spectrometric (ICP-MS) techniques, RIA and ELISAs (Figure 1). Nosustrophine extracts contained saturated- (stearic, 28%; palmitic, 11%) mono- (oleic, 28.1%) and polyunsaturated fatty acids (arachidonic, 14%; docosahexaenoic acid, 14%) of the omega-3, -6 and -9 types, and no trans fatty acids. The analysis further revealed a predominance of the non-essential amino acids glutamic acid (6%), aspartic acid (5.1%), arginine (3.7%) and serine (3.4%), and the essential amino acids leucine (4.4%) and lysine (3.4%) in the extracts. Vitamins B2 (riboflavin), B3 (niacin), D3, and E (tocopherol) were also detected; of these, Vitamin E was the most abundant. The principal nutritional minerals in the extract were calcium (503 mg/kg), magnesium (93 mg/kg), zinc (60 mg/kg) and iron (52 mg/kg). The extract also included a high concentration of neurotransmitters, particularly dopamine (2760.4 pg/mg), L-DOPA (22.5 mg/g), noradrenaline (655.4 pg/mg), serotonin (479 pg/mg), and histamine (158.5 pg/mg). 

To determine the potential roles of Nosustrophine, we next conducted a comparative proteomic analysis with LC-MS/MS using whole extracts of this compound, in collaboration with INIBIC (Appendix A). The UniProt/Swiss-Prot database was then used to identify proteins by searching for each MS/MS spectrum. Since the Swiss-Prot database for *Sus scrofa* only contains 3575 proteins, we incorporated the general mammalian Swiss-Prot database into our search. Consequently, a total of 517 proteins were identified, with a false discovery rate of 1%, including two or more peptides with a confidence of at least 95% and a protein pilot total score ≥ 2. This list included several proteins that are explicitly linked to AD, including Aβ A4, adenosylhomocysteinase (AHCY), apolipoprotein (APO) E, A and C, cathepsin D, choline O-acetyltransferase, neuroendocrine protein 7B2, nicotinamide phosphoribosyltransferase (NAMPT), and presenilin 2 (Table 1). Several proteins implicated in various other AD-related pathways such as neurotransmitter metabolism, inflammation, cerebral oxygenation, oxidative stress, neuroprotection, apoptosis, synapse, and brain metabolism were also identified (Appendix A).

Bioinformatic classification of identified proteins was conducted with the PANTHER Classification System. These findings are presented as pie charts, organized by molecular functions (Figure 1A,B). The primary molecular functions of proteins found in Nosustrophine extracts were related to binding (37.5%) and catalytic activities (41.1%) (Figure 1A). A large number of proteins associated with structural molecular integrity (7.3%), regulation of molecular functions (6%) and transporter activity (4.8%) were also detected (Figure 1A). The main cellular processes regulated by the proteins detected in Nosustrophine were linked to cellular and metabolic processes (43.3%), followed by biological regulation (13.8%), localization (14.2%), and responses to stimuli and signaling (8%) (Figure 1B).

The proteins found in Nosustrophine regulate several cellular pathways, some of them related to neurodegenerative diseases such as Huntington’s disease (actin, calpain and tubulin), AD (presenilins), and Parkinson’s disease (dopamine receptor-mediated signaling pathway with proteins such as aromatic L-amino acid decarboxylase, catechol-O-methyl transferase, monoamine oxidase, protein kinase A and dopamine and cAMP-regulated phosphoproteins) (Figure 1C). A substantial number of proteins in the extract are implicated in chemokine and cytokine-mediated inflammation, integrin signaling, Wnt signaling, cytoskeletal regulation by Rho GTPase and endothelin signaling pathways. (See Appendix A for details on neurodegenerative disease pathways regulated by the proteins identified in Nosustrophine).

### 3.2. Nosustrophine Regulates AD-Related Gene Expression In Vivo

Next, we asked whether Nosustrophine may have therapeutic impact against AD. Our goal was to use a validated preclinical model of AD for studying the effects of Nosustrophine at different stages of the disease [43,44]. APP/BIN1/COPS5 triple-transgenic (3xTg) mice exhibit increased age-related Aβ accumulation and deposition, substantial cell death and neuroinflammation in the cortex and hippocampus [43,44]. 

We took advantage of 3–4- and 8–9-month-old APP/BIN1/COPS5 AD mice to immunolocalize NeuN and TH in the brain (Figure 2). NeuN immunoreactivities in the CA1 subfield (Figure 2G) and DG (Figure 2H) of the hippocampus decreased significantly by 21% (*p* < 0.05) and 41% (*p* < 0.05), respectively, in 3–4-month-old APP/BIN1/COPS5 3xTg-AD mice compared to wild-type controls. Moreover, NeuN-positive staining in the 8–9-month-old 3xTg-AD hippocampus was 41% lower in the CA1 region (*p* < 0.01) and 69% lower in the DG (*p* < 0.05) than in 3–4-month-old transgenic AD animals. In the VTA in the midbrain, TH-immunoreactive neurons were also significantly lower (71%) in 3–4-month-old APP/BIN1/COPS5 mice (Figure 2I) versus wild-type mice (*p* < 0.001). TH immunoreactivity further decreased by 57% in the 8–9-month-old 3xTg-AD VTA compared to 3–4-month-old AD mice (*p* < 0.01), confirming that only dopaminergic neurons were lost in the VTA. This finding was specific to female transgenic mice. Dopamine (DA) dysregulation is reported to be a pathological hallmark in AD patients and is linked to altered cognitive function [45]. VTA degeneration, suggested to be one of the first events in the early (pre-plaque) stages of AD, causes lower DA outflow towards the hippocampus and correlates well with deficits in synaptic plasticity in the CA1 subfield and memory [46,47].

Sparse extracellular Aβ deposition appears in the 0–1-month-old APP/BIN1/COPS5 3xTg-AD mouse cortex and hippocampus, and substantial Aβ deposition occurs in those same brain regions as early as six months of age in these mice [43]. To investigate the effect of Nosustrophine on amyloid pathology, we performed dot blot analysis (Figure 3A–D) in the mouse hippocampus with an antibody highly specific for detecting Aβ1-42, a key driver of Aβ toxicity and amyloid formation [48]. There were no differences in Aβ1-42 immunoreactivity between saline- and Nosustrophine-treated young (3–4-month-old) APP/BIN1/COPS5 mice (Figure 3C), or between saline- and Nosustrophine-treated young wild-type animals. However, in 8–9-month-old APP/BIN1/COPS5 mice, Nosustrophine treatment decreased Aβ1-42 immunoreactivity levels by 39% compared to saline-treated APP/BIN1/COPS5 mice (*p* < 0.05) Figure 3D). 

The expression of several genes associated with neuroinflammation is linked to various neurodegenerative disorders [49]. Since a number of AD-related genes are differentially expressed in APP/BIN1/COPS5 mice compared to wild-type mice [44], we next investigated whether the expression of neurodegenerative-associated genes was altered in the mouse hippocampus in response to treatment with Nosustrophine. We treated young (3–4-month-old) wild-type or APP/BIN1/COPS5 mice with saline or Nosustrophine (*n* = 4 per group) and then analyzed hippocampal presenilin 1 (*PSEN1*), presenilin 2 (*PSEN2*), *APOE*, microtubule-associated protein tau (*MAPT*) and ATP binding cassette subfamily B member 7 (*ABCB7*) expression. Presenilin-1 (PSEN1), one of the four core proteins in the gamma secretase complex, is involved in the generation of Aβ from amyloid precursor protein (APP) [50].

AD patients with an inherited form of the disease may carry mutations in PSEN1 and PSEN2 proteins [42]. *PSEN2* mRNA levels are 70% lower in APP/BIN1/COPS5 mice than in saline-treated wild-type mice [44]. *PSEN1* and *PSEN2* mRNA levels were not altered by Nosustrophine in wild-type animals (Figure 4A). In the hippocampus of Nosustrophine-treated APP/BIN1/COPS5 mice, *PSEN1* expression was elevated (Figure 4A), but *PSEN2* levels were significantly higher (60%, or 2.7-fold) in Nosustrophine-treated- than in saline-treated APP/BIN1/COPS5 mice (Figure 4B), and reached levels similar to wild-type animals. The ε4 allele of apolipoprotein E (APOE) is a major genetic risk factor for sporadic (late-onset) AD [51]. Triple-transgenic AD mice have 65% lower *APOE* mRNA levels than saline-treated wild-type mice [44]. In the current study, Nosustrophine-treated young APP/BIN1/COPS5 mice had approximately three-fold-higher *APOE* levels than saline-treated APP/BIN1/COPS5 mice (Figure 4C); these levels were similar to saline-treated wild-type animals. *MAPT* gene polymorphisms increase AD risk [52]. However, Nosustrophine treatment did not change *MAPT* mRNA levels in any group (Figure 4D). ABC transporters are important contributors to AD [53]. Nosustrophine treatment decreased *ABCB7* expression in wild-type and transgenic mice. However, this decrease was five-fold higher and statistically significant only between Nosustrophine-treated wild-type- and saline-treated wild-type mice (Figure 4E). These findings show that Nosustrophine reduces mRNA levels of several AD-related genes in young animals, and suggest that Nosustrophine is a prophylactic when administered prior to the onset of AD pathology. 

Next, to determine whether Nosustrophine treatment affected AD-related gene expression post-AD-associated neuropathological damage, we analyzed *PSEN1*, *PSEN2*, *APOE*, *MAPT* and *ABCB7* mRNA expression in the 8–9-month-old Nosustrophine- and saline-treated-APP/BIN1/COPS5 mouse hippocampus. Compared to APP/BIN1/COPS5 mice treated with saline, in Nosustrophine-treated mice, *PSEN2* expression was five-fold higher (Figure 5B), *APOE* expression increased eight-fold, and *ABCB7* expression decreased but this was non-significant (Figure 5C). *PSEN1* and *MAPT* expression levels increased, but these were also non-significant (Figure 5A,D). These findings indicate that Nosustrophine regulates *PSEN2* and *APOE* gene expression post AD-related damage, suggesting that Nosustrophine possesses both preventive and therapeutic properties.

### 3.3. Nosustrophine Regulates Inflammation-Related Gene Expression in Transgenic Mice with AD

Inflammation contributes to AD onset and progression [54,55]. Therefore, we next examined the expression of several cytokines and inflammation-related genes in both young (3–4-month-old) and older (8–9-month-old) APP/BIN1/COPS5 and wild-type mice. In younger APP/BIN1/COPS5 mice, *IL-1β* expression was not regulated in APP/BIN1/COPS5 compared to wild-type mice, and Nosustrophine treatment did not affect levels of *IL-1β* (Figure 6A). *IL-6* and *TNFα* mRNA levels are higher in the brain of APP/BIN1/COPS5 than wild-type mice [54]. In the current study, APP/BIN1/COPS5 mice exhibited higher *IL-6* (Figure 6B) and *TNFα* mRNA (Figure 6C) levels than wild-type mice. However, this increase in *TNFα* expression was not statistically significant. Nosustrophine treatment reduced *IL-6* and *TNFα* levels in wild-type and APP/BIN1/COPS5 mice. Nitric oxide synthase 3 (NOS3) and cyclooxygenase-2 (COX-2) are enzymes linked to neuroinflammation [55,56], and *NOS3* and *COX-2* are upregulated in the APP/BIN1/COPS5 mouse brain compared to wild-type mice [44]. In this study, *NOS3* mRNA levels in the APP/BIN1/COPS5 mouse hippocampus were over two-fold higher than in wild-type mice (Figure 6D). Nosustrophine treatment reduced *NOS3* expression in APP/BIN1/COPS5 mice to levels similar to that in wild-type animals. Following the administration of Nosustrophine, *COX-2* expression decreased by more than four-fold in wild-type and transgenic mice brains (Figure 6E). This indicates that Nosustrophine decreases inflammation in a mouse model of AD by regulating inflammation-related gene expression to levels similar to those in wild-type mice. Our findings reveal that Nosustrophine prevents and reduces neuroinflammation.

To determine whether Nosustrophine treatment affected neuroinflammation-related gene expression caused by AD-associated neuropathology, we analyzed *IL-1β*, *IL-6*, *TNFα*, *COX-2* and *NOS3* mRNA levels in older (8–9-month-old) APP/BIN1/COPS5 mice. There were no statistically significant differences in mRNA levels among any of these genes after treatment with Nosustrophine (Figure 7), indicating that this bioproduct does not alter neuroinflammation-related gene expression in older mice. These findings suggest that Nosustrophine does not induce anti-neuroinflammatory effects against existing AD pathology in the brain but is instead preventive against neuroinflammation.

### 3.4. Nosustrophine Regulates DNA Methylation in Trigenic AD Mice

DNA methylation is a biomarker of AD and other neurodegenerative disorders [57,58]. Following Nosustrophine treatment, 5-methylcytosine (5mC) levels increased 1.8-fold in the hippocampus of young (3–4-month-old) wild-type mice and 2.8-fold in young (3–4-month-old) mice APP/BIN1/COPS5 animals (Figure 8A). Nosustrophine treatment did not affect DNA methyltransferase 1 (DNMT1) expression in wild-type and in APP/BIN1/COPS5 mice (Figure 8B). *DNMT3a* expression is lower in the APP/BIN1/COPS5 mouse brain than in controls 57]. In young transgenic 3xTg-AD mice, Nosustrophine significantly increased *DNMT3a* expression more than three-fold versus young saline-treated AD mice (*p* < 0.05; Figure 8C). However, in 8–9-month-old APP/BIN1/COPS5 mice treated with Nosustrophine, *DNMT3a* expression increased in these animals after Nosustrophine treatment but this result was not statistically significant (Figure 8F). Moreover, there were no statistically significant differences in 5mC levels between these older mice (Figure 8D). These findings suggest that Nosustrophine has a strong effect on DNA methylation when taken as a preventive therapy, but has reduced efficacy in the presence of existing AD-related neuropathology. 

### 3.5. Nosustrophine Regulates SIRT Activity

Sirtuins are implicated in several ageing-related biological processes, including the stress response, mitochondrial dysfunction, protein aggregation, oxidative stress, and inflammation [59,60]. Sirtuins, particularly SIRT1, regulate AD-related processes such as APP processing, neuroinflammation and degeneration [60]. SIRT mRNA- and activity levels are lower in APP/BIN1/COPS5 mice versus wild-type animals [44]. In the current study, there was a small, but non-significant, decrease in SIRT activity in wild-type young mice after Nosustrophine treatment (Figure 9A). There were no changes in SIRT activity in young APP/BIN1/COPS5 mice after Nosustrophine treatment (Figure 9A). *SIRT1* mRNA levels were higher in Nosustrophine-treated wild-type and transgenic mice than in saline-treated mice, but these differences were non-significant (Figure 9B). SIRT activity was unaffected in older saline-treated APP/BIN1/COPS5 mice (Figure 9C). However, in older transgenic mice treated with Nosustrophine, *SIRT1* levels increased 2.5-fold (Figure 9D). These findings show that Nosustrophine upregulates *SIRT1* expression in AD mice and suggest that this bioproduct may be used as a natural epidrug to promote SIRT1 transcription in the treatment of AD.

To demonstrate that SIRT1 mRNA upregulation has epigenetic effects in older 3xTg-AD mice after Nosustrophine treatment, we immunoblotted for a chromatin-associated substrate of SIRT1, acetylated-histone H3 (Lys14), in mice hippocampi (Figure 10A,B). Following the administration of Nosustrophine, acetylated-histone H3 protein levels significantly decreased by 24% in Nosustrophine-treated APP/BIN1/COPS5 mice versus saline-treated transgenic animals (*p* < 0.05) (Figure 10A).

### 3.6. Nosustrophine Regulates HDAC Activity

HDAC inhibitors are neuroprotective and increase synaptic plasticity and learning and memory in transgenic mouse models of AD [61]. HDAC activity and *HDAC3* expression are higher in APP/BIN1/COPS5 than in wild-type mice [44]. In the current study, HDAC activity increased in the 3–4-month-old saline-treated APP/BIN1/COPS5 mouse hippocampus (Figure 11A). In comparison, HDAC activity in transgenic mice treated with Nosustrophine decreased by 33%. *HDAC3* expression increased in saline-treated transgenic mice (Figure 11B). By contrast, Nosustrophine-treated 3–4-month-old wild-type and transgenic AD mice exhibited a two-fold reduction in *HDAC3* expression. In older (8–9-month-old) APP/BIN1/COPS5 mice, there were no changes in HDAC activity or *HDAC3* expression between mice injected with saline or Nosustrophine (Figure 11C,D). These findings suggest that Nosustrophine is a preventive treatment for AD by lowering HDAC activity and *HDAC3* expression. However, Nosustrophine has no effect once AD pathology has been established.

## 4. Discussion

In this study, we investigated whether Nosustrophine, a novel bioproduct derived from the porcine brain through non-denaturing biotechnological procedures, is effective for treating AD by targeting key components of the epigenetic machinery, and is more of a preventive measure rather than a therapy for AD. One possible reason for failed AD clinical trials is that they have been conducted on AD patients who had already suffered irreversible neural damage from Aβ toxicity and amyloidogenesis before they were diagnosed and treated. In such patients, neurodegeneration may have progressed to the point where the disease cannot be treated. The earliest loss of neurons in AD occurs a decade or more before the onset of symptoms [62], and is the major reason we had chosen 10–14-month-old mice (equivalent to 38–47-year-old human adults) as this reflects the ages during which the earliest stages of AD begin. Detection and therapeutic intervention early in the AD continuum may therefore be the most effective strategy to reduce the burden of AD and improve the quality of life for patients with the disease. 

Here, through a combination of qPCR and epigenetics-related activity assays, we found that Nosustrophine, compared to vehicle-treated transgenic mice, regulated: AD-related *PSEN2* and *APOE* gene expression, inflammation-related (*NOS3* and *COX-2*) gene expression in 3–4-month-old only, global (5mC)- and de novo DNA methylation (*DNMT3a*), *HDAC3* expression and HDAC activity in 3–4-month-old mice, and Aβ1-42 levels, *SIRT1* expression and acetylated histone H3 protein levels in 8–9-month-old mice (Figure 12A,B). The present study is consistent with our previous report showing that *APOE* and *PSEN2* mRNA levels are reduced in the APP/BIN1/COPS5 mouse model of AD [44]. Nosustrophine treatment increased *PSEN2* and *APOE* expression, suggesting that Nosustrophine is protective against AD. There is no change in *MAPT* expression in the brain of AD patients and controls [58]. Similarly, we found no differences in hippocampal *MAPT* mRNA levels between wild-type and transgenic AD mice. Mass spectrometry analysis of Nosustrophine extracts revealed the presence of APOE and PSEN2 proteins, which may explain the increase in *APOE* and *PSEN2* expression observed in Nosustrophine-treated mice. In fact, the AD-associated presenilin pathway was one of the major signaling pathways identified in Nosustrophine. This bioproduct contains different proteins in that pathway, including APP, APP β-ectodomain, APP intracellular domain, C-terminal APP fragment of 99 amino acids, actin, β amyloid, furin, γ-catenin, and nectins.

Neuroinflammation is critical to AD pathogenesis and its progression. IL-1β and IL-6 are upregulated in the blood and cerebrospinal fluid of AD patients [62,63,64] and in the cerebral cortex of APP/PS1 mice [65]. High IL-1β levels in the AD brain are directly linked to plaque formation and progression and the overexpression of neuronal Aβ precursor- and other plaque-associated proteins [64]. Elevated IL-6 levels are associated with the development of diffuse plaques, which represents the early prodromal stage of plaque formation [62]. TNFα, an important mediator of inflammation, is released by microglia and astrocytes [64] and is a key contributor to AD pathogenesis. Our group recently showed that 9-month-old male APP/BIN1/COPS5 triple-transgenic AD mice express significantly higher levels of IL-6, but not TNFα [44]. In the current study, using APP/BIN1/COPS5 mice, we detected significant differences in IL-6 levels in 3–4-month-old mice only, that is, between saline-versus Nosustrophine-treated wild-type mice, and between saline-treated wild-type versus Nosustrophine-treated transgenic AD animals. As in our previous report [44], TNFα levels were lower in the brain of APP/BIN1/COPS5- than in wild-type mice, but these differences were not statistically significant. Our results in this study show that TNFα and IL-6 mRNA levels were upregulated in 3xTg-AD mice, and that Nosustrophine treatment reduced their expression, suggesting that Nosustrophine regulates cytokine signaling. These data indicate that treatment with Nosustrophine reduces inflammation in wild-type and transgenic mice hippocampi, but only when administered during the early stages of AD. Bioinformatic analysis with PANTHER, performed on our proteomic data, revealed that Nosustrophine contains proteins that mediate inflammatory responses such as calcium/calmodulin-dependent kinase II, cell division cycle protein 42, extracellular matrix proteins, F-actin, G protein-coupled receptor protein Gi, integrins, myosin, protein kinase A, Ras GTP-binding proteins and signal transducers and activators of transcription.

NOS3 overexpression promotes apoptosis and COX-2 is a key mediator of inflammation [66,67,68,69]. COX-2 inhibition has been recommended as a treatment option for AD [70]. Non-steroidal anti-inflammatory drugs that inhibit COX-2 reduce the risk of developing AD in a normal-aging population [71]. We previously reported an increase in *NOS3* expression in APP/BIN1/COPS5 mice 3xTg-AD mice [58]. Since we utilized the same transgenic mouse model of AD in the current study, we decided to analyze its expression in response to Nosustrophine treatment. Here, Nosustrophine reduced *NOS3* and *COX-2* mRNA levels in the hippocampus of only 3–4-month-old transgenic AD mice. It is therefore possible that Nosustrophine may be effective if administered early in the course of AD. In AD, increased *NOS3* expression is linked to cortical neuronal cell death, which may result from enhanced p53- and Bax-mediated apoptosis [67]. Those authors further suggest that high levels of phosphorylated-tau and glycogen synthase kinase-3β (GSK-3β), caused by oxidative stress which impairs mitochondrial function, may accumulate preferentially in the AD brain with a concomitant increase in *NOS3* overexpression. However, there are conflicting findings concerning the role of NOS3 in AD. For example, APP/PS1 mice with partial eNOS deficiency have a higher Aβ plaque burden in the brain compared to APP/PS1 mice at 8 months of age, and partial eNOS deficiency increases APP amyloidogenic processing [72].

Global DNA methylation is downregulated in serum and brain tissue of transgenic mice models of AD and in the hippocampus or blood of AD patients [47,57,58]. Furthermore, the levels of DNA methylation and DNA hydroxymethylation in the hippocampus of AD patients are lower than in those without AD [57,73,74]. Genes encoding DNMT1 and *DNMT3a* contribute to learning and memory [73,75] *NMT* expression is reduced during aging [76] and is decreased in the hippocampus of AD patients [57]. We previously demonstrated that *DNMT1* mRNA levels are not regulated in buffy coat samples from patients with AD [57], whereas *DNMT3a* expression is downregulated in buffy coats from dementia patients and in the brain of saline-treated APP/BIN1/COPS5 AD mice [57]. B-vitamins, resveratrol, catechins, caffeic acid, isoflavones and curcumin, among other natural compounds, regulate DNA methylation in AD [77]. In the current study, Nosustrophine treatment significantly increased global DNA methylation levels in 3–4-month-old APP/BIN1/COPS5 mice only. *DNMT1* expression was not regulated across all experimental groups. *DNMT3a* mRNA levels were higher all Nosustrophine-treated APP/BIN1/COPS5 mice than in saline-treated transgenic mice. However, this difference was statistically significant only when comparing 3–4-month-old saline-treated- and Nosustrophine-treated transgenic mice. These data indicate that Nosustrophine is an epinutraceutical bioproduct that regulates DNA methylation in the APP/BIN1/COPS5 AD mouse hippocampus. PANTHER analysis of the mass spectrometry data obtained from Nosustrophine extracts revealed the presence of AHCY, a regulator of methylation [78]. We did not measure the concentration of AHCY in Nosustrophine extracts in the present study. AHCY interacts with and increases the activity of DNMT1 [79]. During methyl transfer, S-adenosylmethionine (SAM) is converted to S-adenosylhomocysteine (SAH), which is subsequently hydrolyzed by AHCY to Hcy [80]. If Hcy is not metabolized by the trans-sulfuration pathway or re-methylation into methionine, then AHCY would favor SAH biosynthesis over hydrolysis [81]. AHCY does in fact regulate de novo DNA methylation [81]. Thus, the detection of AHCY in Nosustrophine may help explain, at least in part, the increased DNA methylation in mice treated with Nosustrophine (Figure 13A).

Chromatin remodeling and histone post-translational modifications play critical roles in AD. Changes in *SIRT* expression in neurodegenerative disorders [82] and modulation of SIRT1 levels and/or activity is observed in models of aging and neurodegenerative diseases [83,84]. Since *SIRT1* overexpression ameliorates AD [85], increasing SIRT1 enzyme activity is an appealing, potential, therapeutic option for AD [85]. Resveratrol, a naturally occurring chemical found in grapes, peanuts and berries, increases SIRT1 activation more than ten-fold [14,85]. In the U.S., a *Gingko biloba* leaf extract (EGb 761) containing flavone glycosides and proanthocyanadins is promoted as a memory-enhancing supplement [86].

EGb 761 improves cognitive function in patients with mild-to-moderate dementia and in individuals with mild cognitive impairment [87]. EGb 761 protects against β-amyloid-induced neurotoxicity in the N2a neuroblastoma cell line through SIRT1 activation [88]. Administration of polyphenols to APP/PS1 mice increases SIRT1 expression in the brain, blocks the aggregation of tau, modulates APP processing, and reduces memory deficits [85]. Resveratrol activates SIRT1, disrupting β-amyloid aggregation and clearing senile plaques [89]. In the present study, Nosustrophine did not alter SIRT activity. Since the SIRT activity assay used in the current study measures global SIRT activity, it is possible that only SIRT1 activity increased and not the activities of other SIRTs. However, qPCR showed that Nosustrophine increased *SIRT1* mRNA levels in both wild-type and APP/BIN1/COPS5 AD mice. This increase was higher in Nosustrophine-treated APP/BIN1/COPS5 mice than in saline-treated transgenic animals; however, this difference was statistically significant only when comparing 8–9-month-old transgenic mice. These findings indicate that *SIRT1* expression is transcriptionally regulated, and imply that the effects of Nosustrophine on *SIRT1* expression are protective and curative, with the ability to boost *SIRT1* mRNA levels even after AD-related neural damage has occurred. In AD mice, histone acetylation is linked to age-related impairments in learning and memory, and non-selective HDAC inhibitors attenuate cognitive decline in these animals [90]. SIRT1 deacetylates histones H1 (Lys26), H3 (Lys9 and 14), and H4 (Lys16) [91], and deacetylation of other SIRT1 substrates (e.g., p53, FOXO1-FOXO4) is known to protect against neurodegeneration in mouse models of AD [92,93]. In the current study, our Western blot data showed that Nosustrophine promoted deacetylation of histone H3 (Lys14) in hippocampal samples from older APP/BIN1/COPS5 AD mice, indicating that Nosustrophine is protective against AD via the regulation of chromatin-remodeling. The present data, together, provide further evidence that Nosustrophine is an epinutraceutical bioproduct that regulates *SIRT1* expression and the levels of histone H3 acetylation in an in vivo mouse model of AD.

One of the proteins identified following mass spectrometric analysis was NAMPT, a rate-limiting enzyme in the NAD+ biosynthesis pathway; NAMPT overexpression increases SIRT1 activity [94]. We did not measure NAMPT concentrations in Nosustrophine extracts in the current study. In the normal brain, NAMT and NAD+ levels decrease with age [95,96]. Increasing NAD+ levels stimulates vasodilation, blood flow to the brain and cognitive function. In neurodegenerative disorders such as AD, NAMPT-mediated NAD+ biosynthesis is reduced, and NAD+ levels are low [97]. In APP/PS1 transgenic AD mice, NAMPT inhibition increased AD metrics, whereas NAD+ injections attenuated these key indicators of AD [41]. Caloric restriction, exercise and stress increase NAMPT activity, thereby promoting NAD^+^ biosynthesis and activating Sirtuins [96]. That pathway is critical for regulating aging as reduced NAD+ biosynthesis decreases Sirtuin activity, which in turn promotes the development of age-related and neurodegenerative diseases [96]. As a result, increasing NAD+ synthesis and release using NAD+ intermediates and precursors is gaining traction as an effective therapeutic strategy against diseases such as AD [98]. The presence of NAMPT in Nosustrophine may account for the increase in SIRT1 expression in Nosustrophine-treated older mice (Figure 13B).

HDAC inhibitors, first used to treat cancer, improve neuroplasticity and learning and memory in patients with AD and may therefore be neuroprotective against this disease [59]. In clinical trials with AD patients, HDAC inhibitors (sodium butyrate, suberoylanilide hydroxamic acid/Vorinostat, Trichostatin A, and valproate) improve memory, decrease cognitive impairments, and reduce endogenous Aβ production [73]. Omega-3 fatty acids delay cognitive decline in the elderly if taken before the onset of AD symptoms; docosahexaenoic acid decreases HDAC3 levels in neuroblastoma cells [99]. However, in patients already diagnosed with AD, the advantages of omega-3 supplementation appear to be modest [100]. HDAC activity was significantly lower in the hippocampus of 3–4-month-old Nosustrophine-treated APP/BIN1/COPS5 AD mice than in saline-treated transgenic animals. In 3–4-month-old mice only, *HDAC3* expression was significantly lower in Nostrophine-treated wild-type mice compared to saline-treated wild-type mice; a similar reduction in *HDAC3* expression was detected in Nostrophine-treated transgenic mice versus saline-treated transgenic animals. These findings suggest that Nosustrophine is an epinutraceutical bioproduct that functions by downregulating *HDAC3* mRNA levels in vivo, which may help to ameliorate AD-related neurodegeneration.

Since pathology-induced epigenetic modifications are reversible, targeting these alterations may be a novel therapeutic strategy for treating multifactorial diseases including AD. Epidrugs, compounds with targeted action against the epigenome or enzymes with epigenetic activity [101], have shown compelling results in vitro and in animal models for the treatment of different diseases [73,101]. The potential of using a combination of epigenetic drugs in in vitro assays for different cancer treatments has been demonstrated [102]. HDAC inhibitors such as vorinostat, sodium butyrate or suberoylanilide hydroxamic acid suppress the growth of different tumor types in vitro and in vivo [103]. The synergistic combination of the histone inhibitor MS-275 with resveratrol reduces infarct volume and neurological deficits in a mouse model of brain ischemia [104]. Different natural compounds protect against neurodegeneration and contain epinutraceutical properties [14]. The principal polyphenol in green tea (*Camilla sinensis*) is epigallocatechin-3-gallate (EGCG). Treatment with EGCG inhibits fibrillization and improves mitochondrial dysfunction in cultured cell lines [14,105]. EGCG treatment prevents cells death in Aβ-treated rat cortical neuronal cultures [105,106]. Treatment with curcumin protects neurons from oxidation and restores mitochondrial function in animal models of neurotoxicity [107]. Mice fed a high-fat diet exhibit cognitive disturbances and inflammation. In that model, resveratrol reduces pro-inflammatory gene expression, levels of markers of oxidative stress, global methylation and *HDAC2* and *Dnmt3b* expression [108]. The DNMT1 inhibitors azacitidine and decitabine are U.S. FDA-approved for treating acute myelomonocytic leukemia and myelodysplastic syndromes [109]. Our group recently showed that the anti-tumor epinutraceutical bioproduct AntiGan modulates DNA methylation and SIRT activity and expression [110]. DNMT inhibitors may be beneficial for treating multiple sclerosis or scopolamine induced memory deficits in preclinical studies [111,112]. There is currently no approved epidrug for treating neurodegenerative disorders [73]. Our study suggests that Nosustrophine is a promising nutraceutical bioproduct that targets AD-related epigenetic mechanisms.

Various natural compounds such as alkaloids, flavones and flavonoids, curcumin, polyphenols and resveratrol are proposed AD therapeutics [14,108,113]. Clinical trials comparing natural products with conventional anti-dementia medications reveal similar rates of efficacy between traditional medicine and Donepezil-treated patients [114]; however, tolerability is superior with natural products [114]. Nosustrophine and other brain-derived preparations are promising treatments for AD. Cerebrolysin, a porcine brain-derived peptide extract, is efficacious in several clinical trials of AD [115]. Cerebrolysin is reported to contain peptides from digested pig brain or digested porcine brain proteins, as well as neuropeptides, fragments of neurotrophic peptides/factors, and unprocessed neurotrophic factors such as BDNF [28]. However, comprehensive analysis of Cerebrolysin using UHPLC and quadrupole-ion mobility-time-of-flight mass spectrometry (Q-IM-TOFMS) identified its main components to be actin, myelin basic protein, and tubulin α- and β-chains [3]. No fragments of neurotrophic factors such as BDNF, CNTF and NGF are found in Cerebrolysin [28]. By contrast, Nosustrophine is produced through non-denaturing biotechnological procedures; the extract is not subjected to enzymatic or alcoholic lysis, thereby preserving growth factor activity. To this, Nosustrophine extracts contained BDNF and neurotransmitters, mainly dopamine, noradrenaline, and serotonin. There is a correlation between AD and decreased concentrations of dopamine, norepinephrine [116], and serotonin [117].

B-vitamins regulate DNA methylation, and deficiency of methyl nutrients such as B2 or B12 cause disturbances in SAM synthesis [118]. Nosustrophine extracts contained high levels of vitamins B2 and B3, which may explain its effects on DNA methylation in the young 3xTg-AD mouse hippocampus. Vitamins A, C and E are widely recognized for their antioxidant properties. Vitamin D deficiency has been linked to AD pathology [119], and increasing vitamin D intake may lessen the risk for developing AD [100]. However, the epigenetic effects of vitamin D relative to AD have not yet been investigated in-depth. Levels of zinc, a trace mineral, were also elevated in Nosustrophine. Zinc deficiency reduces enzymatic use of methyl groups, altering one-carbon metabolism and, as a result, DNA methylation [100]. Furthermore, the involvement of zinc in histone modification is crucial, as all HDACs, with the exception of HDAC class III members, require zinc for activity [100]. Oxidation may also induce changes in histone acetylation [120]. This is significant within the context of epigenetics because oxidative damage hinders the ability of DNA to function as an optimal substrate for DNMTs, resulting in global hypomethylation [121].

## 5. Conclusions

Although AD is the most important neurodegenerative disease with a worldwide cost that exceeds US $700 billion, only one new medication for AD treatment has been approved in the last 15 years [122]. Our in vivo study, using a transgenic mouse model of AD, sought to address this problem of few and ineffective AD-targeted therapies, from a pharmacoepigenetic point of view. Nosustrophine, a novel bioproduct, regulated AD-related gene expression and functions as an epinutraceutical by modulating DNA methylation and SIRT and HDAC activity and expression. According to our data, we propose that Nosustrophine is a therapeutic compound for the potential prevention and treatment of asymptomatic preclinical AD and early prodromal AD with progressive amyloid deposition. Given that an effective and preventive treatment for AD is generally lacking, and that defective pharmagenes occur in about 85% of patients with AD [122], pharmacoepigenetics should be incorporated into drug development processes to effectively treat this disease. Epigenetic-based therapies may, therefore, be beneficial for preventing and/or treating AD and could promote responses to other AD-medications presently available.

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
