# Peer review of "Nosustrophine: An Epinutraceutical Bioproduct with Effects on DNA Methylation, Histone Acetylation and Sirtuin Expression in Alzheimer’s Disease"

_pharmaceutics, 2022, doi:10.3390/pharmaceutics14112447_

Round 1

Reviewer 1 Report

The manuscript is well written and clear, and easy to follow. The main problems concern the model and the adequacy of the parameters analyzed and that were used to draw the conclusions.

First, the choice of the animal model is not fully explained and the contribution of other genes, in addition to the APP gene, to mimic AD is not presented. The authors test the effect of the Nosustrophine extract in a triple transgenic model that develops Abeta formation, however the effect of this extract on amyloid pathology was not analyzed, rather they choose to analyze several mediators of signaling pathways that are associated with AD but are also altered in other pathologies. In addition, the effect of the extract on behavioral changes was not evaluated, which would be expected for a preventive or therapeutic strategy to be transposed to AD.

The number of animals used per experimental group or their age is not justified. If in terms of life phase equivalencies for mice versus humans, 3–6-month-old and 10–14-month-old mice are considered equivalent to mature-(20–30-year-old) and middle-aged (38– 47-year-old) human adults, it is unclear why 3–4-month-old and 8–9-month-old are referred to as young, and older (30-37-year-old?), respectively. What justifies these ages given the characteristics of the disease?

Conclusions are based on quantification of mRNA levels. At least in those whose levels are altered, the respective protein should be quantified by western blot, as there is not always a direct relationship between mRNA and protein levels.

Another aspect concerns Fig. 2, in which microscopy images are shown but without quantification. Furthermore, why not analyze neurons described as altered in AD, namely glutamatergic or cholinergic? Instead, only dopaminergic neurons were analyzed.

Author Response

Dr Olaia Martínez-Iglesias

Department of Medical Epigenetics

EuroEspes Biomedical Research Center

Bergondo, 15165

Corunna

Spain

Phone:

E-mail: epigenetica@euroespes.com

13 October 2022

Ms. Erna Zhang

Assistant Editor

Pharmaceutics

MDPI

RE: Manuscript ID: pharmaceutics-1924120

Dear Ms. Zhang,

Thank you for giving us the opportunity to submit a revised draft of our manuscript titled “Nosustrophine: an epinutraceutical bioproduct with effects on DNA methylation, histone acetylation and Sirtuin expression in Alzheimer's disease”. We are grateful for the helpful and instructive criticism given by the reviewers and hope that the revised manuscript will be acceptable for publication as a Research Article in the special issue Alzheimer’s Disease and Mild Cognitive Impairment: Emerging Therapeutic Targets and Treatment Strategies in Pharmaceutics.

We have been able to incorporate changes to reflect most of the suggestions provided by the reviewers and have highlighted the changes within the manuscript. The points raised by the reviewers have been addressed as follows (reviewers’ comments are italicized); “Track Changes” has been used to reflect modifications to the revised manuscript.

REVIEWER 1

Comment 1:

The manuscript is well written and clear, and easy to follow. The main problems concern the model and the adequacy of the parameters analyzed and that were used to draw the conclusions.

First, the choice of the animal model is not fully explained and the contribution of other genes, in addition to the APP gene, to mimic AD is not presented. The authors test the effect of the Nosustrophine extract in a triple transgenic model that develops Abeta formation, however the effect of this extract on amyloid pathology was not analyzed, rather they choose to analyze several mediators of signaling pathways that are associated with AD but are also altered in other pathologies. In addition, the effect of the extract on behavioral changes was not evaluated, which would be expected for a preventive or therapeutic strategy to be transposed to AD.

Response:

Thank you to the reviewer. The main objective of this study was to investigate the epigenetic effects of Nosustrophine. We have recently described APP/BIN1/COPS5 3xTg mice as a suitable model for evaluating epigenetic changes in AD, the discovery of new epigenetic-related biomarkers to aid AD diagnosis, and the evaluation of new epidrugs for the treatment of AD (Martinez-Iglesias et al. 2022). 3xTg-AD mice exhibit not only Aβ but also tau pathology (Sterniczuk R et al. 2012), a high level of anxiety and fear (Sterniczuk R et al. 2012), severe neuropathological degeneration, and deficits in synaptic plasticity, object recognition, and learning (Filali M. et al. 2012). In addition, APP/BIN1/COPS5 3xTg-AD mice strongly express AD-related pathological hallmarks (Martinez-Iglesias et al. 2022).

We have addressed the reviewer’s question related to Aβ pathology in Results section “3.2. Nosustrophine regulates AD-related gene expression in vivo” (pages 11 and 12; lines 448–456, and 477–493) in the revised manuscript. To assess the effect of Nosustrophine on amyloid pathology, we performed dot blot analysis in the mouse hippocampus with an antibody specific for identifying Aβ1-42, a main driver of Aβ toxicity and amyloid formation (see new Figures 3A–D).

            We did not conduct behavioral testing in this study because our major goal was to investigate the role of the epigenetic machinery and other biomarkers in response to Nosustrophine treatment in 3xTg-AD mice. Nonetheless, we do agree that behavioral studies are important and have begun to conduct these experiments. Unfortunately, we did not have time to incorporate those data in the revised manuscript but have included this as a limitation of the study (Section 6, page 24).

References

Martínez-Iglesias, O., Naidoo, V., Carrera, I., & Cacabelos, R. (2022). Epigenetic Studies in the Male APP/BIN1/COPS5 Triple-Transgenic Mouse Model of Alzheimer's Disease. International journal of molecular sciences, 23(5), 2446.

Sterniczuk R.; Dyck R.H.; Laferla F.; Antle M.C. (2010) Characterization of the 3xTg-AD mouse model of Alzheimer´s disease. Brain Res, 12, 139-148.

            Filali, M., Lalonde, R., Theriault, P., Julien, C., Calon, F., & Planel, E. (2012). Cognitive and non-cognitive behaviors in the triple transgenic mouse model of Alzheimer's disease expressing mutated APP, PS1, and Mapt (3xTg-AD). Behavioural Brain Research, 234(2), 334–342

Comment 2:

The number of animals used per experimental group or their age is not justified. If in terms of life phase equivalencies for mice versus humans, 3–6-month-old and 10–14-month-old mice are considered equivalent to mature-(20–30-year-old) and middle-aged (38– 47-year-old) human adults, it is unclear why 3–4-month-old and 8–9-month-old are referred to as young, and older (30-37-year-old?), respectively. What justifies these ages given the characteristics of the disease?

Response:

Thank you to the reviewer. In this study, we used four animals per group. We have previously worked with this mouse model of AD and have established that 3–5 mice per group is a good n number for obtaining statistically significant results.

In terms of life phase equivalencies for mice versus humans, the earliest loss of neurons in AD occurs a decade or more before symptoms appear. In patients with severe AD, the disease may have progressed to the point where neural damage is irreversible and cannot be treated. This is the reason we had selected 10–14-month-old mice (equivalent to 38–47-year-old human adults) as they represent the ages during which AD has started and is progressing, but without severe neuropathology. Early detection and treatment intervention may therefore be the most successful strategy for reducing the burden of AD and improving the quality of life for persons with the disease. This information has been added to the Discussion section (paragraph 1, page 19) in the revised manuscript.

Comment 3:

Conclusions are based on quantification of mRNA levels. At least in those whose levels are altered, the respective protein should be quantified by western blot, as there is not always a direct relationship between mRNA and protein levels.

Response:

Thank you to the reviewer. Our aim in this study was to examine the mRNA levels of several important markers. We focused on the epigenetic effects in the AD mouse brain in response to Nosustrophine treatment, and gene expression studies were only to demonstrate the effects of Nosustrophine on AD pathology. We therefore do not have any data on their protein expression at the current time as this was outside the aim and scope of our study. We will, furthermore, need to procure the corresponding antibodies, and given the limited timeframe for manuscript resubmission, it is not possible for us to presently conduct Western blot studies. We appreciate this comment, however, and agree that these are important experiments to perform. Our future objective is to investigate their protein expression with Western blotting and ELISAs.

Comment 4:

Another aspect concerns Fig. 2, in which microscopy images are shown but without quantification. Furthermore, why not analyze neurons described as altered in AD, namely glutamatergic or cholinergic? Instead, only dopaminergic neurons were analyzed.

Response:

Thank you to the reviewer for pointing this out. In the revised manuscript, we have now quantified neuronal nuclear protein (NeuN) immunostaining in the CA1 subfield (new Figure 2G) and dentate gyrus (new Figure 2H) in the hippocampus, and tyrosine hydroxylase (TH) immunoreactivity in the ventral tegmental area (VTA) in the midbrain (new Figure 2I) in wild-type controls versus 3xTg-AD mice (3–4- and 8–9-month-old APP/BIN1/COPS5 mice).

            We chose to analyze dopaminergic neurons because alterations in the dopaminergic (DAergic) system are frequently reported in patients with AD (Nobili A et al. 2017). Those authors showed that an age-dependent loss of DAergic neurons occurs in the VTA at pre-plaque stages, but that TH-positive neurons in the substantia nigra pars compacta (SNpc) DAergic neurons were unaffected. We have clarified in this revised manuscript that this selective degeneration of DAergic neurond in the VTA causes lower DA outflow in the hippocampus and nucleus accumbens (NAc). DAergic cell death progression is associated with aberrant CA1 synaptic plasticity, memory function, and food-reward processing. In this APP/BIN1/COPS5 mouse model of AD, DAergic neuronal death in the VTA at pre-plaque stages may contribute to memory deficits. Early dopaminergic dysfunction has been previously described in 3xTg-AD (APPswe/TauP301L/PS1M146V+/-) mice with amyloid and tau pathology (Gloria Y et al. 2021). As the 3xTg-AD mice (APP/BIN1/COPS5) used in our study also exhibit Aβ and tau pathology, we analyzed dopaminergic neuron immunoreactivity as well. However, in future studies, we will study glutamatergic or cholinergic contributions in this mouse model of AD.

References

Nobili A.; Latagliata E.C.; Viscomi M.T.; Cavallucci V. et al. (2017) Dopamine neuronal loss contributes to memory and reward dysfunction in a model of Alzheimer's disease. Nat Commun. 8, 14727.

Gloria Y.; Ceyzériat K.; Tsartsalis S.; Millet P.; Tournier B.B. (2021) Dopaminergic dysfunction in the 3xTg-AD mice model of Alzheimer’s disease. Sci Reports, 11, 19412.

We sincerely hope that we have addressed the comments to the satisfaction of the reviewers.

Sincerely,

Dr Olaia Martínez-Iglesias

Department of Medical Epigenetics

EuroEspes Biomedical Research Center

Bergondo, 15165

Corunna, Spain

E-mail: epigenetica@euroespes.com

Reviewer 2 Report

The manuscript entitled “Nosustrophine: an epinutraceutical bioproduct with effects on  DNA methylation, histone acetylation and Sirtuin expression  in Alzheimer's disease” describes the protective effect  of Nosustrophine, a porcine brain derived extract in an animal model of Alzheimer’s disease at early (3-4 months) or older age (8-9 months). The manuscript is well-written, and they performed a complete and careful analysis of the composition using appropriate methodology. In addition, they analyzed whether the treatment modifies DNA methylation, histone modification (acetylation), inflammation and neurodegenerative related gene expression at both ages.

Though the number of animals is a bit low (n=4/group), they obtained interesting results and attempted to make a correlation between the pathways potentially activated by the main components of the extracts and the experimental data. However, the study has some flaws. The only demonstrated epigenetic effect is the increase in 5-methylcytosine in animals treated with Nosustrophine. I think they should investigate a bit more in depth the SIRT1 pathway since the corresponding mRNA is clearly induced by the epinutraceutical bioproduct in old animals. Regarding inflammation, there is some misinterpretation.

L299. Statistical analysis: why do authors use t-test? 2-way ANOVA is most appropriate for 2 factors (genotype and treatment).

L465. In Results, 3.3, I think there is a confusion regarding NOS and inflammation. Nos2 (iNOS), the inducible form of NOS is clearly involved in inflammation and expressed by glial cells and also by endothelial cells. Nos3 (eNOS) is expressed by endothelial cells and plays a critical role in the maintenance of the homeostasis of the brain vasculature. Authors should clarify why they decided to analyze the expression of Nos3 and why they associate it to inflammation. They should also include a discussion regarding Nos3 relevance in AD models since Ahmed et al showed that reduced levels of eNOS increased cognitive deficit (Int J Mol Sci 23 (13):7316 Partial Endothelial Nitric Oxide Synthase Deficiency Exacerbates Cognitive Deficit and Amyloid Pathology in the APPswe/PS1ΔE9 Mouse Model of Alzheimer's Disease). In the present manuscript, authors conclude that the reduction of NOS3 expression is beneficial, whereas Ahmed et al suggest the opposite. What is the rationale to conclude that NOS3 reduction is beneficial in the present manuscript?

L507. Results 3.4. DNA methylation:

It was not clear to me how authors measured the DNA methylation and how they calculated the percentage of 5-methylcytosine. I was unable to find the protocol to measure it. An explanation should be included in the manuscript.

L514: not correct, DNMT3a is NOT statistically downregulated in the Tg animal.

L531. Results 3.5. SIRT activity

In a previous paper from the group (Martinez-Iglesias, 2022, IJMS), authors found a decreased Sirt1 expression in the APP/BIN1/COPS5 transgenic animal, also used in the present study.

In the present manuscript, authors observed an upregulation of the SIRT1 mRNA in old animals after Nosustrophine treatment. Regarding total SIRT activity, there were no changes. Since authors claim that Nosustrophine has epigenetic effects, they should provide evidence of it. I suggest to look at the effect of an inhibitor of SIRT1 (for example Ex527) on SIRT activity in the colorimetric assay to unveil whether increased levels of SIRT1 transcripts are associated with increased activity. In addition, authors should look at the acetylation of a specific substrate of Sirt1 such as p65 of NF-kB to unravel whether the anti-inflammatory effect of Nosustrophine (reduction of Cox2 mRNA) could be related to SIRT1 (see previous section).

Regarding,eNOS, another substrate of SIRT1, even though protein levels of eNOS may be lower after the treatment with Nosustrophine, eNOS maybe deacetylated by SIRT1 and exert a protective effect on the brain vasculature as described in Hattori et al (Silent information regulator 2 homolog 1 counters cerebral hypoperfusion injury by deacetylating endothelial nitric oxide synthase. Stroke 2014; 45:3403–3411).

So, authors should perform a western blot and analyze eNOS levels, Acetylated-eNOS, p65 and Acetylated-p65 in old animals if possible.

However, if authors consider that other SIRT1 substrates are more appropriate than p65 and eNOS in the context of Alzheimer’s disease, such as Acetylated-Tau, or histone3 in relation to chromatin remodeling, they can present data with these substrates.

Minor comments:

Line 455: preventive instead of preventative

L628: “One of the most important pathways identified in Nosustrophine was inflammation driven by chemokine and cytokine signaling”: what are the data supporting this conclusion?

Author Response

Dr Olaia Martínez-Iglesias

Department of Medical Epigenetics

EuroEspes Biomedical Research Center

Bergondo, 15165

Corunna

Spain

Phone:

E-mail: epigenetica@euroespes.com

13 October 2022

Ms. Erna Zhang

Assistant Editor

Pharmaceutics

MDPI

RE: Manuscript ID: pharmaceutics-1924120

Dear Ms. Zhang,

Thank you for giving us the opportunity to submit a revised draft of our manuscript titled “Nosustrophine: an epinutraceutical bioproduct with effects on DNA methylation, histone acetylation and Sirtuin expression in Alzheimer's disease”. We are grateful for the helpful and instructive criticism given by the reviewers and hope that the revised manuscript will be acceptable for publication as a Research Article in the special issue Alzheimer’s Disease and Mild Cognitive Impairment: Emerging Therapeutic Targets and Treatment Strategies in Pharmaceutics.

We have been able to incorporate changes to reflect most of the suggestions provided by the reviewers and have highlighted the changes within the manuscript. The points raised by the reviewers have been addressed as follows (reviewers’ comments are italicized); “Track Changes” has been used to reflect modifications to the revised manuscript.

REVIEWER 2

Major comments

Comment 1:

The manuscript entitled “Nosustrophine: an epinutraceutical bioproduct with effects on DNA methylation, histone acetylation and Sirtuin expression in Alzheimer's disease” describes the protective effect of Nosustrophine, a porcine brain derived extract in an animal model of Alzheimer’s disease at early (3-4 months) or older age (8-9 months). The manuscript is well-written, and they performed a complete and careful analysis of the composition using appropriate methodology. In addition, they analyzed whether the treatment modifies DNA methylation, histone modification (acetylation), inflammation and neurodegenerative related gene expression at both ages.

Though the number of animals is a bit low (n=4/group), they obtained interesting results and attempted to make a correlation between the pathways potentially activated by the main components of the extracts and the experimental data. However, the study has some flaws. The only demonstrated epigenetic effect is the increase in 5-methylcytosine in animals treated with Nosustrophine. I think they should investigate a bit more in depth the SIRT1 pathway since the corresponding mRNA is clearly induced by the epinutraceutical bioproduct in old animals. Regarding inflammation, there is some misinterpretation.

Response:

Thank you to the reviewer. We have addressed the questions in Reviewer Comment 1 in our responses to Comments 3, 6 and 7 below.

Comment 2:

L299. Statistical analysis: why do authors use t-test? 2-way ANOVA is most appropriate for 2 factors (genotype and treatment).

Response:

Thank you to the reviewer for the recommendation. In the revised manuscript, we have now changed the statistical analysis to a one-way ANOVA (young animals) when we have more than two groups, with the Tukey’s post-hoc test.

Comment 3:

L465. In Results, 3.3, I think there is a confusion regarding NOS and inflammation. Nos2 (iNOS), the inducible form of NOS is clearly involved in inflammation and expressed by glial cells and also by endothelial cells. Nos3 (eNOS) is expressed by endothelial cells and plays a critical role in the maintenance of the homeostasis of the brain vasculature. Authors should clarify why they decided to analyze the expression of Nos3 and why they associate it to inflammation. They should also include a discussion regarding Nos3 relevance in AD models since Ahmed et al showed that reduced levels of eNOS increased cognitive deficit (Int J Mol Sci 23 (13):7316 Partial Endothelial Nitric Oxide Synthase Deficiency Exacerbates Cognitive Deficit and Amyloid Pathology in the APPswe/PS1ΔE9 Mouse Model of Alzheimer's Disease). In the present manuscript, authors conclude that the reduction of NOS3 expression is beneficial, whereas Ahmed et al suggest the opposite. What is the rationale to conclude that NOS3 reduction is beneficial in the present manuscript?

Response:

Thank you to the reviewer for this helpful comment. We have previously reported an increase in NOS3 expression in APP/BIN1/COPS5 mice 3xTg-AD mice (Martinez-Iglesias O et al, 2022). Since we used the same transgenic mouse model of AD in the present study, we therefore decided to analyze its expression in response to Nosustrophine treatment. Increased nitric oxide synthase 3 (NOS3) expression is linked to cortical neuronal cell death in AD (de la Monte S.M. et al 2003; de la Monte S.M. et al, 2007). Those authors suggest that increased NOS3 expression in AD may cause neuronal death via enhanced p53- and Bax-mediated apoptosis, and that high phospho-tau and GSK-3β levels (caused by oxidative stress which impairs mitochondrial function) may accumulate preferentially in the AD brain with a concomitant increase in NOS3 overexpression. However, we do acknowledge that there are conflicting findings concerning the role of NOS3 in AD. Ahmed et al. (2022), for example, showed that APP/PS1 mice with partial eNOS deficiency have a higher Aβ plaque burden in the brain compared to APP/PS1 mice at 8-months of age, and that partial eNOS deficiency increased APP amyloidogenic processing. This information has now been added to the Discussion section (paragraph 2, page 20) in the revised manuscript.

References

Ahmed S, Jing Y, Mockett BG, Zhang H, Abraham WC, Liu P. Partial Endothelial Nitric Oxide Synthase Deficiency Exacerbates Cognitive Deficit and Amyloid Pathology in the APPswe/PS1ΔE9 Mouse Model of Alzheimer's Disease. Int J Mol Sci. 2022, 23(13):7316.

de la Monte, S. M., Jhaveri, A., Maron, B. A., & Wands, J. R. (2007). Nitric oxide synthase 3-mediated neurodegeneration after intracerebral gene delivery. Journal of neuropathology and experimental neurology, 66(4), 272–283.

de la Monte, S. M., Chiche, J., von dem Bussche, A., Sanyal, S., Lahousse, S. A., Janssens, S. P., & Bloch, K. D. (2003). Nitric oxide synthase-3 overexpression causes apoptosis and impairs neuronal mitochondrial function: relevance to Alzheimer's-type neurodegeneration. Laboratory investigation; a journal of technical methods and pathology, 83(2), 287–298.

Comment 4:

L507. Results 3.4. DNA methylation:

It was not clear to me how authors measured the DNA methylation and how they calculated the percentage of 5-methylcytosine. I was unable to find the protocol to measure it. An explanation should be included in the manuscript.

Response:

The authors apologize for the oversight. This information has now been included in new section “2.9. Quantification of Global DNA Methylation (5mC)” under Materials and Methods in the revised manuscript. Briefly, DNA was extracted from mice hippocampi with the Qiagen DNA Mini Kit (Qiagen), and only DNA samples with 260/280 and 260/230 ratios greater than 1.8. were used. Global 5mC levels were determined colorimetrically (in 100 ng DNA) using the MethylFlash Global DNA Methylation kit (Epigentek, New York, NY, USA), with the absorbance measured at 450 nm. As per the manufacturer’s guidelines, 5mC levels are expressed as mean (%) ± S.E.M. To quantify the absolute amount of methylated DNA, we generated a standard curve using linear regression (Microsoft Excel). The amount and percentage of 5mC was then calculated with the following formulae:

5mC (ng) = (Sample OD − Blank OD)/(Slope × 2) (1)

5mC (%) = 5mC (ng)/sample DNA (ng) × 100 (2)

Comment 5:

L514: not correct, DNMT3a is NOT statistically downregulated in the Tg animal.

Response:

Thank you to the reviewer. This information has been corrected accordingly in Results section 3.4 (paragraph 1, page 16) in the revised manuscript.

Comment 6:

L531. Results 3.5. SIRT activity

In a previous paper from the group (Martinez-Iglesias, 2022, IJMS), authors found a decreased Sirt1 expression in the APP/BIN1/COPS5 transgenic animal, also used in the present study.

In the present manuscript, authors observed an upregulation of the SIRT1 mRNA in old animals after Nosustrophine treatment. Regarding total SIRT activity, there were no changes. Since authors claim that Nosustrophine has epigenetic effects, they should provide evidence of it. I suggest to look at the effect of an inhibitor of SIRT1 (for example Ex527) on SIRT activity in the colorimetric assay to unveil whether increased levels of SIRT1 transcripts are associated with increased activity. In addition, authors should look at the acetylation of a specific substrate of Sirt1 such as p65 of NF-kB to unravel whether the anti-inflammatory effect of Nosustrophine (reduction of Cox2 mRNA) could be related to SIRT1 (see previous section).

Regarding,eNOS, another substrate of SIRT1, even though protein levels of eNOS may be lower after the treatment with Nosustrophine, eNOS maybe deacetylated by SIRT1 and exert a protective effect on the brain vasculature as described in Hattori et al (Silent information regulator 2 homolog 1 counters cerebral hypoperfusion injury by deacetylating endothelial nitric oxide synthase. Stroke 2014; 45:3403–3411).

Response:

Thank you to the reviewer for these helpful comments. With respect to SIRT1 activity, measurements of SIRT activity measured global levels of different sirtuins rather than just the activity of a specific sirtuin (e.g., SIRT1). This information has been added to the Discussion section (bottom of page 22).

Nonetheless, we wish to confirm that we had followed the recommendation of the reviewer and immediately purchased two antibodies (Abcam) with the intention of Western blotting for NF-kappaB p65 acetyl (K31) and total NF-kappaB p65. Unfortunately, at the time of resubmission of the revised manuscript, those antibodies have not yet been delivered. We agree with the author that this would be an important experiment to perform, and our future goal is to investigate the acetylation of p65 NF-kappaB relative to Nosustrophine treatment in AD mice. However, we did perform SDS-PAGE and Western blotting and examined the protein expression of another substrate of SIRT1: acetylated histone H3 (please see our response to Comment 7 below).

Comment 7:

So, authors should perform a western blot and analyze eNOS levels, Acetylated-eNOS, p65 and Acetylated-p65 in old animals if possible.

However, if authors consider that other SIRT1 substrates are more appropriate than p65 and eNOS in the context of Alzheimer’s disease, such as Acetylated-Tau, or histone3 in relation to chromatin remodeling, they can present data with these substrates.

Response:

We appreciate this insightful comment by the reviewer. As mentioned above in our response to Comment 6, we performed a Western blot to investigate whether Nosustrophine promotes deacetylation of histone H3, a SIRT1 substrate, in older 3xTg-AD mice. Following the administration of Nosustrophine, acetylated-histone H3 protein levels significantly decreased by 24% in Nosustrophine-treated APP/BIN1/COPS5 mice versus saline-treated transgenic animals (p < 0.05) (new Figures 10A, B). In relation to this finding, we have included in the revised manuscript a discussion of the functional relationship between SIRT1 and AD (page 23, Discussion section, lines 830–846).

Minor comments

Comment 1:

Line 455: preventive instead of preventative

Response:

The authors apologize for the error and have corrected this accordingly.

Comment 2:

L628: “One of the most important pathways identified in Nosustrophine was inflammation driven by chemokine and cytokine signaling”: what are the data supporting this conclusion?.

Response:

Our data show that TNF-alpha and IL-6 mRNA levels are upregulated in 3xTg-AD mice, and that Nosustrophine treatment reduces their expression. These data suggest that Nosustrophine regulates cytokine signaling. PANTHER analysis derived from our proteomic studies revealed that Nosustrophine contains proteins that mediate inflammatory responses such as calcium/calmodulin-dependent kinase II, cell division cycle protein 42, extracellular matrix proteins, F-actin, G protein-coupled receptor protein Gi, integrins, myosin, protein kinase A, Ras GTP-binding proteins and signal transducers and activators of transcription (Supplementary Table S1).

We sincerely hope that we have addressed the comments to the satisfaction of the reviewers.

Sincerely,

Dr Olaia Martínez-Iglesias

Department of Medical Epigenetics

EuroEspes Biomedical Research Center

Bergondo, 15165

Corunna, Spain

E-mail: epigenetica@euroespes.com

Round 2

Reviewer 1 Report

The authors have addressed appropriately the concerns of the reviewer

Author Response

REVIEWER 1

Response:

Thank you to the reviewer.

Reviewer 2 Report

The authors addressed all my comments and took into consideration my recommendations. Therefore, I consider they globally gave a satisfactory answer to my concerns. I think it is critical to check for the involvement of SIRT1 in the protective effect of the treatment as they plan to do.

Please change essay for assay line 821.

Author Response

Dr Olaia Martínez-Iglesias

Department of Medical Epigenetics

EuroEspes Biomedical Research Center

Bergondo, 15165

Corunna

Spain

Phone:

E-mail: epigenetica@euroespes.com

03 November 2022

Ms. Zora Zhou

Assistant Editor

Pharmaceutics

MDPI

RE: Manuscript ID: pharmaceutics-1924120

Dear Ms. Zhou,

Thank you for giving us the opportunity to submit a revised draft of our manuscript titled “Nosustrophine: an epinutraceutical bioproduct with effects on DNA methylation, histone acetylation and Sirtuin expression in Alzheimer's disease”. We hope that the revised manuscript will be acceptable for publication as a Research Article in the special issue Alzheimer’s Disease and Mild Cognitive Impairment: Emerging Therapeutic Targets and Treatment Strategies in Pharmaceutics.

We have incorporated changes to reflect the comments by  Reviewer 2. “Track Changes” has been used to reflect modifications to the revised manuscript.

REVIEWER 2

Comment:

Please change essay for assay line 821.

Response:

Thank you to the reviewer. The authors apologize for the oversight. “Essay” has now been changed to “assay” in the revised manuscript” (line 825).

We sincerely hope that we have addressed the comments to the satisfaction of  the reviewer.

Sincerely,

Dr Olaia Martínez-Iglesias

Department of Medical Epigenetics

EuroEspes Biomedical Research Center

Bergondo, 15165

Corunna, Spain

E-mail: epigenetica@euroespes.com